# Airway epithelial specific deletion of Jun-N-terminal kinase 1 attenuates pulmonary fibrosis in two independent mouse models

**Jos L. van der Velden**[1], **John F. Alcorn**[2], **David G. Chapman**[3], **Lennart K. A. Lundblad**[3], **Charles G. Irvin**[3], **Roger J. Davis**[4], **Kelly Butnor**[1], **Yvonne M. W. Janssen-Heininger**[1] *

**1** Department of Pathology and Laboratory Medicine, University of Vermont, Burlington, Vermont, United States of America, **2** Children's Hospital of Pittsburgh University of Pittsburgh Medical Center, Pittsburgh, Pennsylvania, United States of America, **3** Departments of Medicine, University of Vermont, Burlington, Vermont, United States of America, **4** Howard Hughes Medical Institute, University of Massachusetts Medical School, Worcester, Massachusetts, United States of America

* yvonne.janssen@uvm.edu

**Data Availability Statement:** All relevant data are within the manuscript and its Supporting Information files.

## Abstract

The stress-induced kinase, c-Jun-N-terminal kinase 1 (JNK1) has previously been implicated in the pathogenesis of lung fibrosis. However, the exact cell type(s) wherein JNK1 exerts its pro-fibrotic role(s) remained enigmatic. Herein we demonstrate prominent activation of JNK in bronchial epithelia using the mouse models of bleomycin- or AdTGFβ1-induced fibrosis. Furthermore, in lung tissues of patients with idiopathic pulmonary fibrosis (IPF), active JNK was observed in various regions including type I and type II pneumocytes and fibroblasts. No JNK activity was observed in adjacent normal tissue or in normal control tissue. To address the role of epithelial JNK1, we ablated *Jnk1* form bronchiolar and alveolar type II epithelial cells using CCSP-directed Cre recombinase-mediated ablation of LoxP-flanked *Jnk1* alleles. Our results demonstrate that ablation of *Jnk1* from airway epithelia resulted in a strong protection from bleomycin- or adenovirus expressing active transforming growth factor beta-1 (AdTGFβ1)-induced fibrosis. Ablation of the *Jnk1* allele at a time when collagen increases were already present showed a reversal of existing increases in collagen content. Epithelial *Jnk1* ablation resulted in attenuation of mesenchymal genes and proteins in lung tissue and preserved expression of epithelial genes. Collectively, these data suggest that epithelial JNK1 contributes to the pathogenesis of pulmonary fibrosis. Given the presence of active JNK in lungs from patients with IPF, targeting JNK1 in airway epithelia may represent a potential treatment strategy to combat this devastating disease.

## Introduction

Organ fibrosis has been estimated to contribute to 45% of all causes of mortality [1] and contributes to kidney, liver, and heart failure. Idiopathic Pulmonary Fibrosis (IPF) is a devastating disease that kills approximately 40,000 people in the USA each year, with a survival time of 3–5 years upon diagnosis. The causes of IPF remain incompletely understood, but it is thought

**Funding:** This work was funded by NIH R01 HL079331, R01 HL060014, and R35 HL135828 (Y J-H). Analysis of biological specimens provided by the Lung Tissue Research Consortium (LTRC) was supported by R03 HL095404 (Y J-H). The funders had no role in study design, data collection and analysis, decision to publish, or preparation of the manuscript.

**Competing interests:** The authors have read the journal's policy and have the following competing interests: Jos van der Velden and Yvonne Janssen-Heininger received $120000 from Celgene Corporation to test the impact of CC930 and CC90001 on house dust mite-induced fibrotic airway remodeling in mice. There are no patents, products in development or marketed products associated with this research to declare. This does not alter our adherence to PLOS ONE policies on sharing data and materials.

that the disease manifests in setting of aging, as a result of repeated and diverse injuries to the lung. The exact biochemical and cellular pathways that culminate in excessive extracellular matrix deposition and alveolar remodeling are a topic of intense investigation, and encompass death of, or injury to, airway epithelial cells and a failure to regenerate alveolar epithelia, altered immune cell activation, with an alternative, M2-polarized macrophage signature, and excessive accumulation of myofibroblasts. Activated myofibroblasts are thought to be the predominant source of production of excessive extracellular matrix, although collagen secretion by alveolar epithelial cell also has been shown to contribute to fibrosis in mouse models [2, 3].

The role of epithelial cells in pulmonary fibrosis has clearly emerged. Certain patients with familial IPF have mutations in *SpA* and *SpC* genes, and accumulation of misfolded proteins in alveolar epithelial cells has been shown to contribute to endoplasmic reticulum stress [4] and enhanced susceptibility to bleomycin-induced injury [5] and fibrosis [6].The mucin gene, mucin 5B (*Muc5b*) expressed in distal airways, also has been strongly linked to pulmonary fibrosis [7]. The induction of epithelial cell death using an antibody that activates the death receptor, Fas (CD95) [8, 9], or transgenic approaches encompassing diphtheria toxin, or diphtheria toxin receptor targeting bronchiolar or type II epithelial cells [10], are sufficient to induce pulmonary fibrosis in rodents. Furthermore, activation of a partial epithelial to mesenchymal transition (EMT) phenotype in epithelial cells and resultant production of collagen [2] may also constitute a source of ECM in pulmonary fibrosis. While partial EMT is believed to be important to allow repair following acute injury, repeated activation of partial EMT-like responses in epithelial cells may contribute to the pathogenesis of fibrosis through the release of ECM proteins by epithelial cells, without providing a source of fibroblasts [11]. Lastly, altered progenitor function and/or plasticity of epithelial cells also has been associated with IPF, based on the emergence of distinct populations of epithelial cells in patients with IPF characterized by unique RNA expression profiles that suggest intermediate populations, including cells with basal cell characteristics [12]. Lineage negative epithelial cells (LNEP) expressing keratin-5 and p63 [13, 14] also contribute to enhanced populations of epithelial cells in bleomycin-injured lungs, suggesting that airway epithelial cells contribute to alveolar remodeling and/or repair.

c-Jun-N-terminal kinase (JNK), also known as stress activated protein kinase, as its name implies, can be induced by a wide variety of stresses. The JNK pathway is a prominent regulator of cell death, and in some settings is a major driver of apoptosis. Our laboratory has previously shown that JNK1 (but not JNK2) is a major regulator of TGFβ-induced epithelial to mesenchymal transition [15], in association with phosphorylation of SMAD3 [16]. JNK1 also promotes Wnt-3-induced EMT via regulation of beta-catenin [17]. Mice that globally lack *Jnk1* (but not *Jnk2*) were protected from bleomycin-induced fibrosis, or fibrotic remodeling in a model of allergic airways disease [18]. Most recently we demonstrated that JNK1 contributes to production of ECM proteins in basal cells stimulated with TGFβ1, notably of laminin and fibronectin [19]. We furthermore showed that JNK1 contributes to augmentation of an array of mesenchymal genes in basal cells plated on a provisional ECM derived from TGFβ1-stimulated basal cells, and that the provisional ECM activated JNK1 in a Rho Kinase-dependent manner.

Despite these earlier observations, which collectively implicated a role of epithelial JNK1, the role of JNK1 within airway epithelial cells in the pathogenesis of pulmonary fibrosis remains unknown. Using bleomycin and adenovirus expressing active transforming growth factor beta-1 (TGFβ1) in mice, we show prominent activation of JNK in bronchial epithelia, as well as parenchymal regions. In order to unravel the impact of epithelial JNK1 on the pathogenesis of fibrosis, we therefore ablated *Jnk1* in airway epithelial cells, using a club cell

secretory protein (CCSP) promoter (also known as secretoglobin family 1A member 1, SCGB1A1) that targeted both proximal (CCSP-expressing) and distal epithelial (type 2) epithelial cells.

## Materials and methods

### Chemicals and antibodies

All chemicals utilized were purchased from Sigma-Aldrich (St. Louis, MO) unless otherwise noted. Antibodies: Total JNK (#9252), phospho-JNK (#9251) antibodies were obtained from Cell Signaling Technology (Danvers, MA) (all anti Rabbit, 1:1000), antibodies for S100a4 (FSP1) (sc-19949, anti-Goat, 1:500) and actin (sc-8432, anti-mouse 1:5000) were from Santa Cruz Biotechnology (Santa Cruz, CA), antibodies for CCSP (anti goat 1:4000) were a kind gift from Dr. B. Stripp, Cedars-Sinai Medical Center, Los Angeles, CA).

### Human lung tissues

Paraffin-embedded lung tissue samples from IPF patients (n = 5, 3 males and 2 females) and non-diseased controls (n = 5, 3 males and 2 females) were obtained from the National Heart Lung and Blood Institute-sponsored Lung Tissue Research Consortium (LTRC). The clinical data and specimens have been de-identified by the LTRC. LTRC protocols were approved by the Institutional Review Board Committee on Human Research in Medical Sciences (CHRMS # M14-469) at the University of Vermont.

### Phospho-JNK staining human lung tissues

Paraffin sections were cut from formalin-fixed and paraffin-embedded tissue. pJNK was detected using a polyclonal antibody against rabbit pJNK (#44-682G, Invitrogen, Carlsbad, CA, USA). pJNK staining was performed on lung sections after antigen retrieval by incubation of slides for 20 minutes in 0.01M sodium citrate pH 6.0 at 95ºC. Slides were then blocked with 2% normal goat serum for 30 minutes, followed by incubation with the antibody against p-JNK (1:250 dilution) overnight at 4ºC. After application of biotin-conjugated swine anti-rabbit IgG Ab (DakoCytomation, Glostrup, Denmark) and alkaline phosphatase-labeled avidin–biotin complex (Vector, Burlingame, CA, USA), enzymatic reactivity was visualized using the Vector Blue Substrate Kit (Vector). Sections were counterstained with Nuclear Fast Red (Vector) and mounted. Pictures were taken with a Leica VERSA whole slide image scanner at 400x magnification (Purchased with the aid of a University of Vermont College of Medicine Shared Instrumentation Award)

### Animals

Mice with a germ-line mutation in the *Jnk1* gene with LoxP elements inserted into two different introns (*Jnk1*LoxP) [20] were crossed with CCSP-rtTA and TetO-Cre bi-transgenic mice, in order to create *CCSP*-rtTA/tetO-Cre/ *Jnk1*loxP/loxP genotype. CCSP-rtTA and TetO-Cre transgenic mice were a gift from Dr. Whitsett (University of Cincinnati College of Medicine, Cincinnati, OH)[21]. The first generation CCSP-rtTA transgene was used in the present studies. For all experiments, age-matched male and female mice were used and all experimental mice where 8–10 weeks old at the initiation of the studies. CCSP-rtTA/tetO-Cre/*Jnk*1 loxP/loxP were fed doxycycline (Dox) containing chow (6 g/kg TestDiet, Richmond, IN) in order to delete *JNK*1 in CCSP and SPC positive cells (ΔEpi *Jnk*1). Control mice were age-matched transgene-negative littermates (WT), or bi-transgenic mice expressing CCSP-rtTA/tetO-Cre transgenenes along with the *JNK*1 WT allele (WT-*Jnk*1) and were fed dox to control for potential off

target effects of dox, or expression of the rtTA tetO-Cre transgenes [21]. An additional control consisted of mice consisting all three CCSP-rtTA/tetO-Cre/*Jnk*1[loxP/loxP] transgenes, in which dox-food was withheld (*Jnk*1 loxP). In order to confirm ablation of JNK1 from lung epithelial cells following administration of doxycycline (dox), single cell suspensions were prepared for flow cytometric evaluation as described previously [22]. The EpCAM positive, CD45 negative, Sca1-low fraction of lung cells was sorted (representing epithelial cells), and JNK1 content was assessed in this fraction and compared to the CD31/34/45[+] fraction, representing other non-epithelial lung cell types. Mice were housed with a 12-h:12-h light/dark cycle and allowed free access to standard laboratory chow and water. The Institutional Animal Care and Use Committee at the University of Vermont approved all animal studies. Experiments were repeated at least once and numbers of animals in each study are indicated in each Figure Legend.

## Mouse models of fibrosis

Bleomycin (5U/kg body weight) or 5X10[8] PFU recombinant AdTGFβ1223/225 (provided by Dr. Jack Gauldie, McMaster University) was administered oropharyngeally, as described previously [18, 23]. PBS (Bleomycin) or Ad5 adenovirus (adTGFβ1) (Vector Biolabs, Philadelphia, PA), were administered as respective controls. Fibrosis was evaluated in mice 3 weeks post administration of bleomycin or AdTGFβ1. In selected experiments doxycycline was administered 7 days before the treatment of profibrotic agents. Mice were maintained on dox food until the completion of the experiment. In the delayed ablation regimens, dox food was withheld until 14 days after administration of AdTGFβ1 and thereafter mice were maintained on Dox food until completion of the experiment. Schematics depicting the time courses of JNK1 ablation, pro-fibrotic agent exposure and euthanization are included in S1 Fig (S1 Fig).

## Assessment of fibrosis

Lung sections were stained with Masson's trichrome reagent to stain collagen, causing a blue staining of collagen. Slides were scored using a scale of 0 to 3 (0 being the least stain intensity, 3 the highest intensity) for collagen deposition by two independent, blinded investigators. The cumulative score from each mouse was then averaged according to treatment group. Total lung collagen was measured in the right middle lobe of the lung after overnight digestion with 10 mg/ml pepsin in 0.5 M acetic acid using the Sircol Assay (Biocolor, UK) as directed by the manufacturer. As an independent measure of fibrosis, we also measured hydroxyproline content in the right superior lung lobe [24].

## Assessment of airway mechanics

Mice were anesthetized with intraperitoneal pentobarbital sodium (90mg/kg), tracheotomized, and mechanically ventilated at 200 breaths/min with a tidal volume of 0.25 ml and positive end-expiratory pressure of 3cmH$_2$O (FlexiVent; SCIREQ, Montreal, Quebec, Canada). Tissue elastance (H) was measured as a measure of the stiffness of the lung [25, 26].

## Homogenization of cell and lung tissue and Western blotting

Protein lysates were prepared by harvesting cells or mincing lung tissue in cold lysis buffer immediately followed by homogenization as previously described [16, 18]. Lysates were incubated on ice for 30 min, followed by 30 min of centrifugation at 16,000 g. A portion of the supernatant was saved for protein determination, before the addition of Laemmli sample buffer. Total protein was assessed by the Bio-Rad DC Protein Assay kit (Bio-Rad). Total JNK1/2, phospho JNK1/2, and β-actin protein abundance were evaluated by Western blotting. Two

**Table 1. Primer sequences used in real time PCR analyses.**

| Gene | Accession | | Sequences (5′ → 3′) | Amplicon (bp) |
|---|---|---|---|---|
| *Hmga2* | NM_010441 | forward | aaggcagcaaaaacaagagc | 121 |
| | | reverse | gcaggcttcttctgaacgac | |
| *Acta2 (α-SMA)* | NM_007392.2 | forward | ctgacagaggcaccactgaa | 160 |
| | | reverse | catctccagagtccagcaca | |
| *S100a4* | NM_0113111 | forward | ctggggaaaaggacagatga | 109 |
| | | reverse | tgcaggacaggaagacacag | |
| *Cdh1 (E-cadherin)* | NM_009864.2 | forward | agccattgccaagtacatcc | 133 |
| | | reverse | aaagaccggctgggtaaact | |
| *Col1A1* | NM_007742.3 | forward | gagcggagagtactggatcg | 103 |
| | | reverse | gttcgggctgatgtaccagt | |
| *Vim (Vimentin)* | NM_011701.4 | forward | tgaaggaagagatggctcgt | 100 |
| | | reverse | tccagcagcttcctgtaggt | |
| *Scgb1a1(CCSP)* | NM_011681.2 | forward | atctgcccaggatttcttca | 150 |
| | | reverse | ctcttgtgggagggtatcca | |
| *Tjp1 (ZO-1)* | NM_009386.2 | forward | ccacctctgtccagctcttc | 249 |
| | | reverse | caccggagtgatggttttct | |
| *Sftpc (SPC)* | NM_011359.2 | forward | cagctccaggaacctactgc | 121 |
| | | reverse | agcttagaggtgggtgtgga | |

dominant bands of JNK immunoreactivity are detected in Western blots, each of which contain both JNK1 and JNK2 which are not fully resolved using conventional Western blotting procedures. The top band preferentially contains JNK2 while the lower band preferentially contains JNK1 [16], as is indicated in the figures.

## Gene expression

Total RNA was isolated from mouse lungs, using the RNeasy mini-kit (Qiagen, Valencia, CA) as directed by the manufacturer. Isolated RNA was subjected to reverse transcription and DNase treatment to produce cDNA for Taqman gene analysis using SYBR green. PCR data were analyzed by using the ΔΔCt method of relative quantification. Primer sequences were taken from GeneBank. All accession numbers are listed in Table 1.

## Statistical analysis

Data were evaluated by one-way ANOVA using the Tukey test to adjust for multiple comparisons. Results with a P-value less than 0.05 or smaller were considered statistically significant. Nonparametric exact permutation tests were conducted using SPSS Version 19.0.

## Results

### Increases in pJNK immunoreactivity in airway epithelial cells in fibrotic lung tissues

We first sought to demonstrate that phosphorylation of JNK, indicative of its activation, occurred in lung tissue using two different mouse models of fibrotic lung remodeling, bleomycin, and adenovirus expressing active transforming growth factor beta 1 (AdTGFβ1, Fig 1A and 1B). Administration of either bleomycin or AdTGFβ1 in mice results in both parenchymal and subepithelial fibrosis [18]. In mice exposed to AdTGFβ1 or bleomycin, phosphorylation of JNK (pJNK) was increased in homogenized lung tissue (Fig 1A). Of note, two prominent

**A**

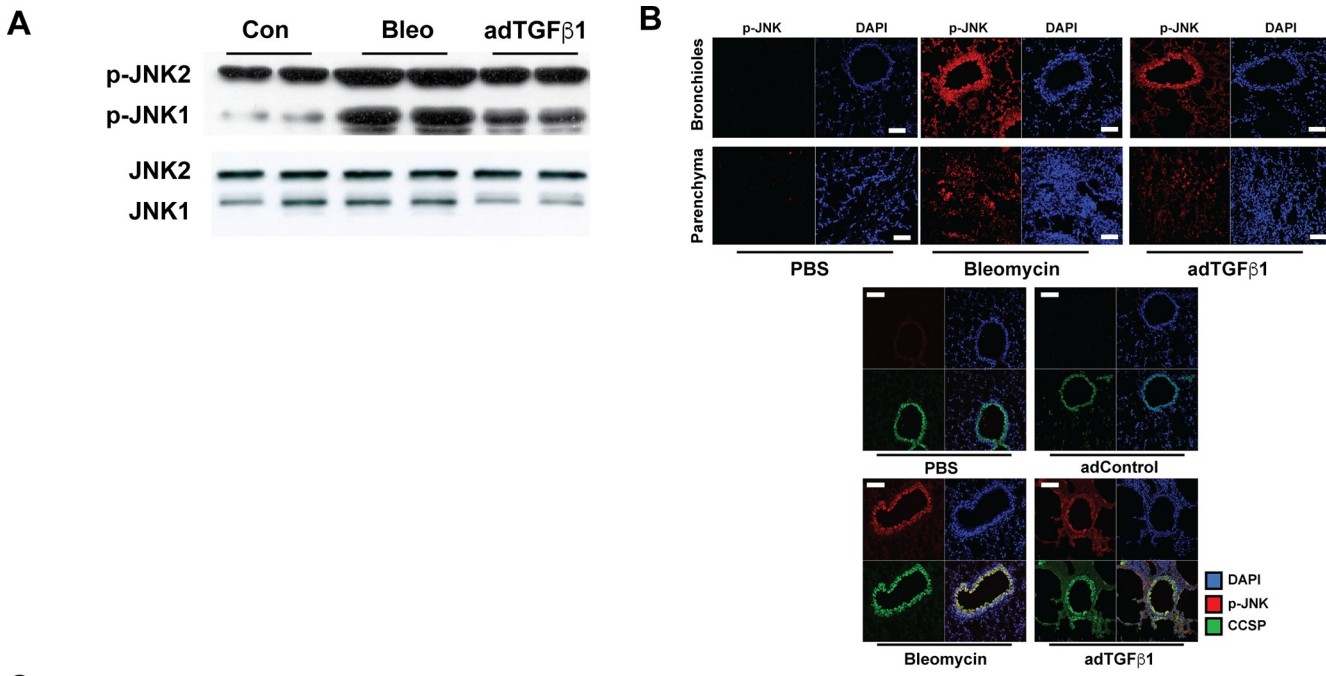

**B**

**C**

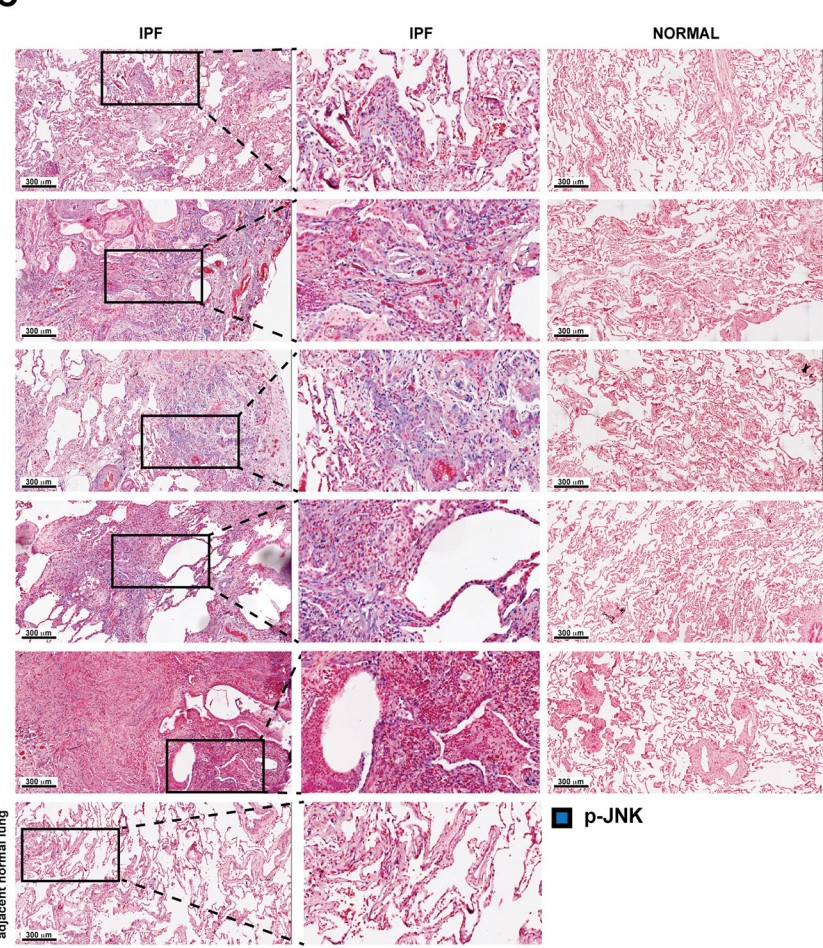

**Fig 1. Phospho-JNK immunoreactivity is increased in lung tissues in settings of airways and parenchymal fibrosis.** (**A**) p-JNK in lung tissue derived from mice with bleomycin or adTGFβ1-induced pulmonary fibrosis. Mice were exposed to agents, and respective vehicle controls (con), and 3 weeks thereafter, lung tissues were homogenized for the assessment of pJNK via Western blot analysis. PBS or control-adenoviral vector exposed mice resulted in similar lack of p-JNK immunoreactivity (see Fig 1B below), and therefore only one representative control group (Con) is shown. (**B**) Immunolocalization of p-JNK (red) in lung tissue from mice 3 weeks after exposure to a representative vehicle control, bleomycin or adenovirus expressing recombinant active transforming growth factor beta-1 (adTGFβ1). Lower Panels; Co-localization of p-JNK with CCSP in control vehicle, bleomycin or adTGFβ1. Results were evaluated via confocal microscopy. Co-localization of p-JNK and CCSP is indicated by a yellow color. Scale bars: 50 μm. (**C**) p-JNK (blue) in lung tissue from patients with idiopathic pulmonary fibrosis (IPF) or non-IPF lung (Normal, n = 5 patients/group). Scale bars: 300 μm.

bands were detected via Western blot analysis, each representing splice variants of JNK1 and JNK2, the top band mainly representing JNK2 (JNK2/1), while the lower band mainly represents JNK1 (JNK1/2). Immuno-reactivity of the lower JNK band was more prominently increased compared to the upper band, consistent with activation of JNK1. Confocal microscopy analysis demonstrated strong increases in pJNK immunofluorescence both within airway epithelium and parenchymal regions, the latter potentially reflecting type II epithelial cells. Dual staining of p-JNK and CCSP indeed demonstrated strong co-localization of p-JNK and CCSP in mice with AdTGFβ1- or bleomycin-induced fibrosis. (Fig 1B). In order to corroborate these observations, we next evaluated pJNK immuno-localization in 5 different patients with idiopathic pulmonary disease (IPF) as well as 5 non-IPF lungs (normal). Evaluation of the staining by a board-certified pulmonary pathologist revealed that pJNK staining was readily apparent in multiple cell types, including epithelial cells (type I and II alveolar pneumocytes) in patients with IPF (Fig 1C). pJNK reactivity was also observed in fibroblasts and inflammatory cells in IPF lung. Little or no staining was found in normal control tissue or in adjacent normal lung tissue from IPF patients. Overall, these data demonstrate increases in pJNK in epithelial cells in two different models of fibrotic airway and parenchymal remodeling, as well as in areas of remodeling in patients with IPF.

## Epithelial-specific ablation of JNK1 protects against fibrotic remodeling

In order to conclusively determine the contribution of epithelial JNK1 in fibrotic remodeling, we selectively ablated JNK1 from bronchiolar (CCSP-expressing) epithelial cells and from type II cells (Fig 2A–2C). We chose this approach due to the prominent reactivity of JNK in CCSP expressing cells, and to ensure targeting of distal epithelial cells, given the reactivity of JNK in parenchymal regions (Fig 1). Following administration of doxycycline (dox), JNK1 was selectively ablated from the epithelial fraction of lung cells (ΔEpi *Jnk1*, Fig 2B).

We next administered dox to mice one week prior to oropharyngeal administration of bleomycin (Fig 3A–3E). Ablation of JNK1 from airway epithelial cells resulted in an almost complete protection from the development of bleomycin-induced fibrosis, as evidenced by the strong diminution of hydroxyproline (Fig 3A), masson's trichrome-reactive material (Fig 3B), collagen content, measured with picrosirius red reagent (Fig 3C) and quantification of masson's trichrome reactivity (Fig 3D). While bleomycin led to increases in tissue elastance in WT mice, these increases were not observed in mice lacking epithelial JNK1 (Fig 3E). We next addressed whether a similar protective effect of ablation of epithelial JNK1 occurred in the AdTGFβ1-model of fibrosis given our prior observations that JNK1 augments TGFβ1-signaling in airway epithelial cells [16]. Administration of dox one week prior to AdTGFβ1 also almost completely prevented AdTGFβ1-mediated increases in hydroxyproline content (Fig 4A) Masson's trichrome reactive material (Fig 4B), lung collagen content (Fig 4C), and increases in the masson's trichrome score (Fig 4D), similar to our observations in the bleomycin model. In the absence of dox-containing food, mice containing the *Jnk1*$^{LoxP/LoxP}$, *CCSP*-rtTA, and TetO-Cre alleles showed increases in fibrosis in response to AdTGFβ1 or bleomycin

**A**

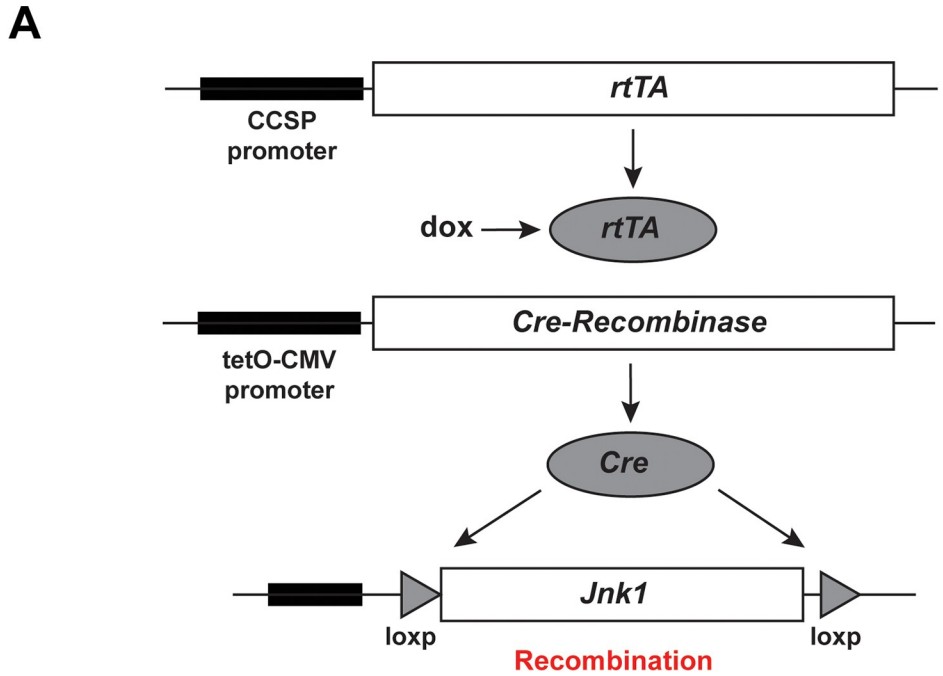

**B**

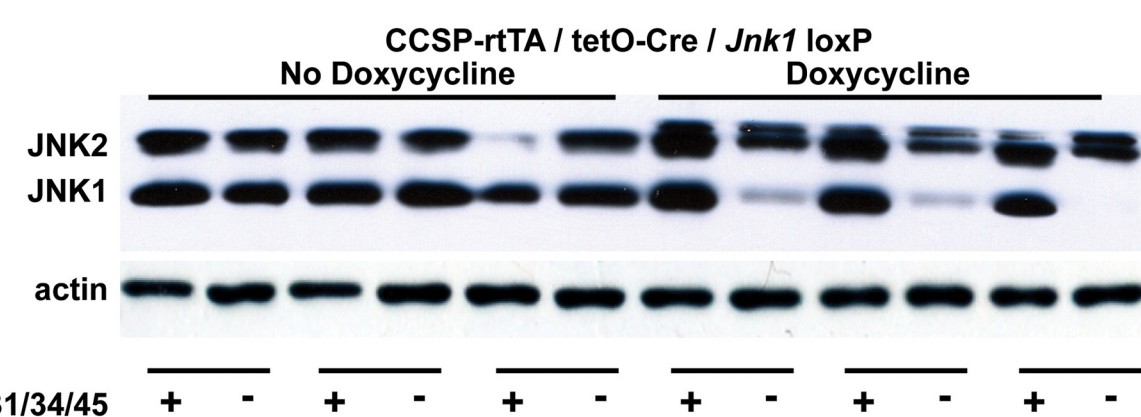

**C**

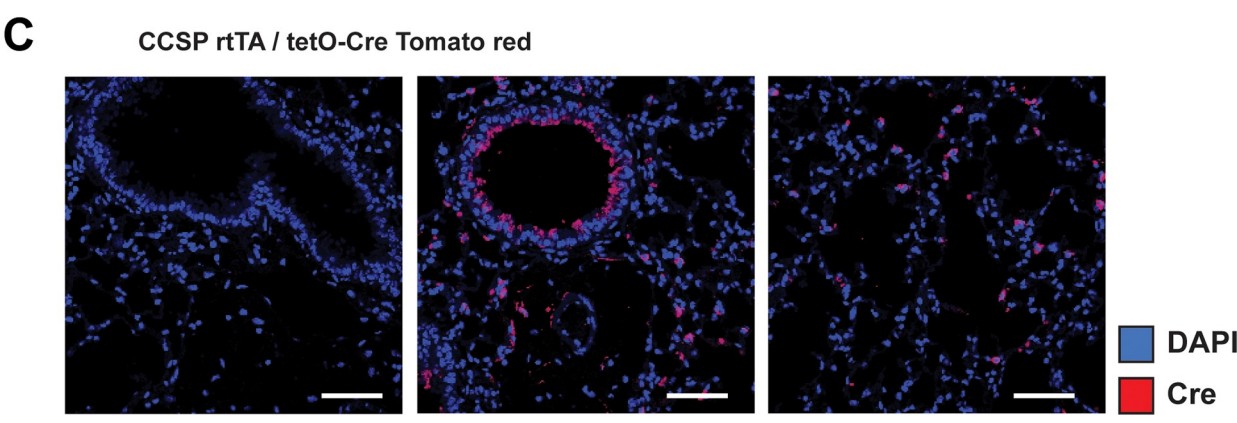

**Fig 2. Depiction of the conditional ablation strategy of JNK1 from lung epithelial cells.** (**A**) Schematic of JNK1 ablation from airway epithelial cells using transgenic mice expressing CCSP-rtTA, TetO-Cre, and *Jnk1*<sup>loxP/loxP</sup> alleles. Upon feeding doxycycline (dox)- containing food, JNK1 is ablated from airway epithelial cells, designated as ΔEpi *Jnk1*. (**B**) Confirmation of ablation of JNK1 in airway epithelial cells. 3 weeks after initiation of Dox feeding, single cells lung suspensions were created and cells were sorted by flow cytometry. The EpCAM positive, CD45 negative, Sca1 low fraction of lung cells was isolated for assessment of JNK1 expression via Western Blot analysis. Note that JNK1 was only ablated from the EpCAM positive, CD45 negative, Sca1low fraction of lung cells. (**C**) Visualization of Cre-recombination in CCSP-rtTA, TetO-Cre-expressing mice which were bred with Tomato red reporter mice. Red: Cre-expressing cells. Scale bars: 50 μm.

that were indistinguishable from WT transgene negative animals (Fig 4E). Overall, these findings demonstrate that ablation of JNK1 in airway epithelia provides protect against the development of fibrosis using two independent models.

We next addressed the impact of epithelial ablation of JNK1 on expression profiles of epithelial and mesenchymal genes and mesenchymal proteins in homogenized lung tissues from bleomycin- or AdTGFβ1-exposed mice (Figs 5 and 6, respectively). As expected, increases in expression of mesenchymal genes or proteins and decreases in epithelial genes were observed in lung tissues from WT animals subjected to bleomycin (Fig 5A–5C). Importantly the increased mesenchymal profile induced by bleomycin was almost completely prevented in mice with epithelial specific ablation of JNK1 (Fig 5A and 5C). Conversely, decreases in mRNA expression of the epithelial markers *Scgb1a1* (CCSP), *Cdh1* and *Tjp-1* observed in WT mice in response to bleomycin, were ameliorated in mice lacking epithelial JNK1 (Fig 5B). We corroborated these findings using AdTGFβ1, and showed a similar diminution of mesenchymal genes and proteins, following ablation of epithelial JNK1, along with a protection from decreases in expression of the epithelial genes,) *Cdh1* and *Tjp-1* (Fig 6A–6C). These findings implicate epithelial JNK1 as a critical contributor in the development of fibrosis, in association with attenuation of epithelial gene expression profiles, while promoting a mesenchymal expression signature.

## Delayed epithelial specific ablation of JNK1 protects against fibrotic remodeling

We next addressed whether ablation of epithelial JNK1 plays a role in the progression of fibrosis. We therefore first administered AdTGFβ1, and 2 weeks thereafter verified that collagen was increased in lung tissue (Fig 7A and 7B). At that time, we initiated ablation of epithelial *JNK1* with dox-containing food, and assessed collagen deposition 4 weeks thereafter (Fig 7A–7C). AdTGFβ1-mediated increases in collagen deposition were almost completely reversed upon delayed ablation of *JNK1* from airway epithelium (Δ-epi-JNK1) (Fig 7A–7C). In the absence of dox, collagen content increased slightly between weeks 2 and 6 in mice carrying the *JNK1*<sup>LoxP/LoxP</sup>, CCSP-rtTA, and TetO-Cre transgenes (JNK1loxP, Fig 7A). Furthermore, dox feeding of mice carrying CCSP-rtTA, and TetO-Cre transgenes but WT *JNK1* alleles (WT-*JNK1*) developed fibrosis similar to transgene negative WT littermates (Fig 7C). Similar to earlier observations, expression of the mesenchymal genes, *Col1a1*, and *Acta2* (α-SMA) was strongly decreased while *Cdh1* expression was restored to control levels following delayed ablation of epithelial *Jnk1*, compared to WT mice subjected to AdTGFβ1 (Fig 7D). Collectively, these findings demonstrate that presence of JNK1 within the airway epithelium at a time when increases in collagen deposition are already apparent, plays a crucial role in sustaining fibrotic airway remodeling in association with repressed epithelial, and enhanced mesenchymal expression profiles.

## Conclusion

Pulmonary fibrosis has been associated with the activation of diverse signaling pathways that affect multiple cell types. The JNK pathway, notably JNK1, had previously been implicated in

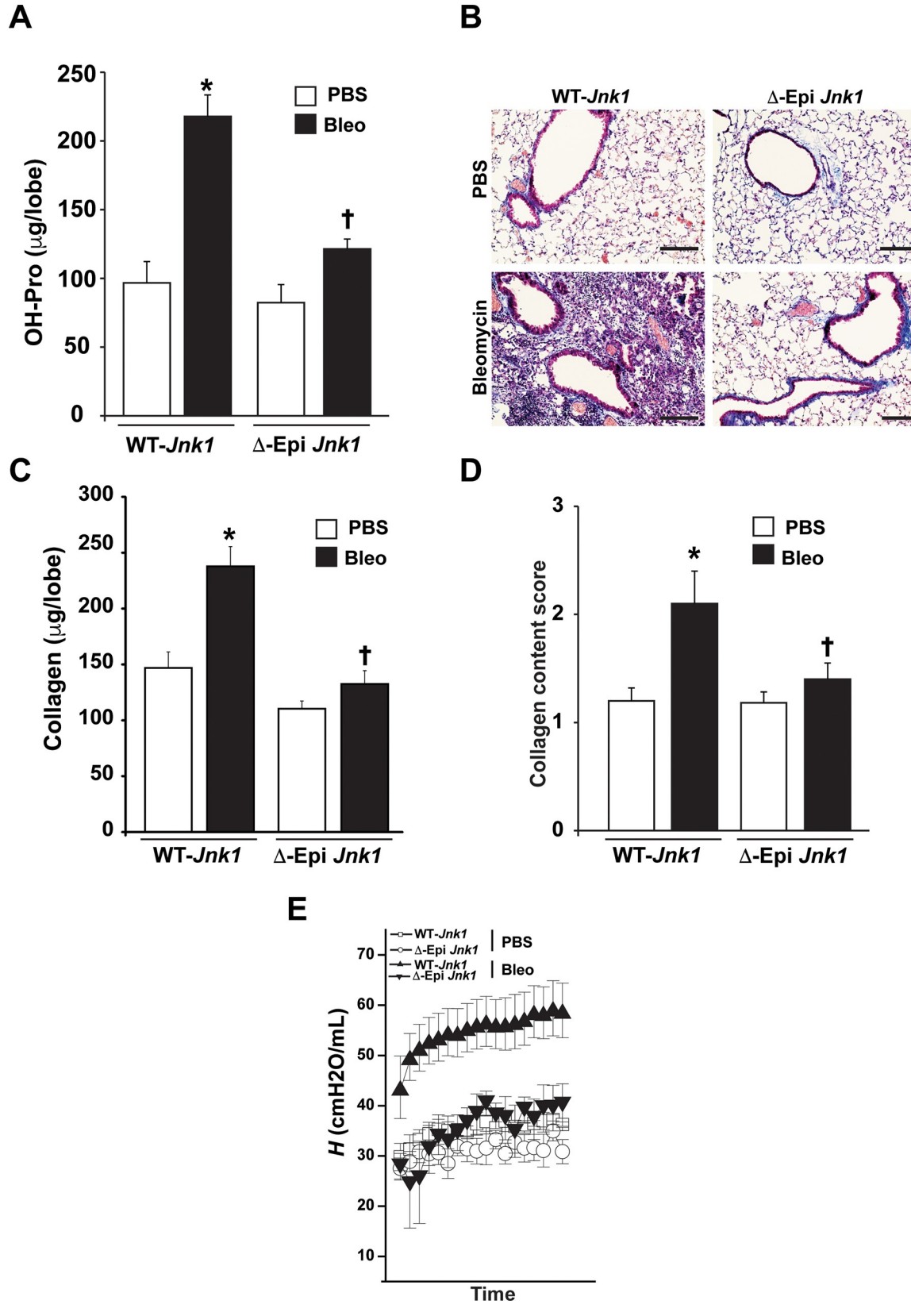

**Fig 3. Conditional ablation of JNK1 from lung epithelial cells attenuates lung fibrosis induced by bleomycin.** JNK1 was selectively ablated from the bronchiolar epithelial cells and type II cells using a CCSP promoter driving a reverse tetracycline activator (rtTA), the tetracyclin operon driving CRE recombinase (TetOP-Cre), and mice carrying a LoxP-flanked *Jnk1* allele. Assessment of total collagen content in lung tissue 3 weeks following administration of bleomycin via hydroxyproline content (**A**), Masson's trichrome staining (**B**), Sircol assay (**C**) and semi-quantitative evaluation of Masson's trichrome reactive material (**D**). (**E**) Assessment of tissue elastance in mice exposed to bleomycin, and the impact of ablation of epithelial JNK1. ΔEpi *Jnk1* mice or respective control groups, as described in the methods section were fed dox food for one week prior to exposure to bleomycin. 3 weeks post-administration of bleomycin, mice were evaluated via forced oscillation mechanics to assess tissue elastance. (WT- *Jnk1*: PBS n = 6, Bleo n = 12, ΔEpi *Jnk1*: PBS n = 5, Bleo n = 8 respectively mice/group from 2 independent experiments). * p< 0.05 compared to PBS control group. † p< 0.05 compared to respective WT groups. (ANOVA). Scale bars: 50 μm.

the pathogenesis of pulmonary fibrosis, based upon observations that mice globally lacking *JNK1*, but not *JNK2*, were protected from the development of bleomycin-induced fibrosis. Deletion of JNK1 had no impact on recruitment of inflammatory cells observed after administration of bleomycin [18]. The exact cell types wherein JNK activation occurs and contributes to disease pathogenesis remained unclear as JNK activation both in epithelial cells [27] as well as fibroblasts [28] had been suggested as potential contributors to the pathogenesis of fibrosis. As was stated above, JNK1 had been linked to the epithelial mesenchymal transition induced by TGFβ1 or Wnt3a [15, 17]. Similarly, JNK1 also had been implicated in myofibroblast activation [29]. We demonstrated herein that in mouse models of fibrosis, JNK activation occurred in bronchial epithelial cells, as well as parenchymal regions within the lung. JNK activation was also observed in type I and II epithelial cells, fibroblast and inflammatory cells in lung tissues from patients with IPF. We furthermore demonstrated that ablation of JNK1 in bronchial and type II epithelial cells, using a CCSP promoter to express Cre Recombinase to remove the loxP-flanked *Jnk1* allele, affords strong protection against fibrosis induced by bleomycin or AdTGFβ1. Significantly, epithelial ablation of JNK1 at a time where collagen was already increased resulted in reversal of AdTGFβ1-induced fibrosis.

Altered plasticity of epithelial cells and failure to regenerate the alveolar epithelium consisting of type II and type I epithelial cells is an important feature of pulmonary fibrosis, with reactive type II epithelial cells and areas of bronchiolarization occurring in lungs from patients with IPF [27]. RNA profiling in individual epithelial cells from patients with IPF identified multiple distinct epithelial gene signatures, with features of mixed basal and alveolar characteristics, indicative of altered plasticity, and inadequate alveolar repair [30]. Results from our current studies fail to pin point the exact location wherein JNK1 is operative to contribute to fibrosis, given that we used the a CCSP expression cassette that is expressed both in bronchiolar epithelial cells and distal epithelia, likely representing type II cells [21]. It also is not clear how JNK1 mechanistically controls pathways that govern epithelial plasticity. JNK1 can regulate both Smad and Wnt-signaling pathways, which has been implicated in epithelial cell fate, and epithelial plasticity [16, 17, 31]. Additional studies will be required to elucidate whether Wnt is a pro-fibrotic target of JNK1.

We recently showed that JNK1 activity in airway basal cells contributes to their mesenchymal activation, indicative of partial EMT, when these cells were plated on a provisional matrix derived from TGFβ1-stimulated basal cells, or on a fibronectin or laminin matrix [19]. Basal cells plated onto a decellularized lung scaffold prepared from control mice showed no evidence of JNK activation and retained expression of p63 and keratin 5. Interestingly, these cells also begun to express CCSP, indicative of their differentiation into secretory cells. In contrast, basal cells plated onto a decellularized scaffold prepared from lungs of mice with bleomycin-induced fibrosis showed JNK activation, enhanced expression of mesenchymal genes and proteins, and loss of epithelial genes along with decreases in expression of p63 and keratin 5. Basal cells lacking JNK1 were resistant to this loss of "epithelial-ness", and similar protective effects were

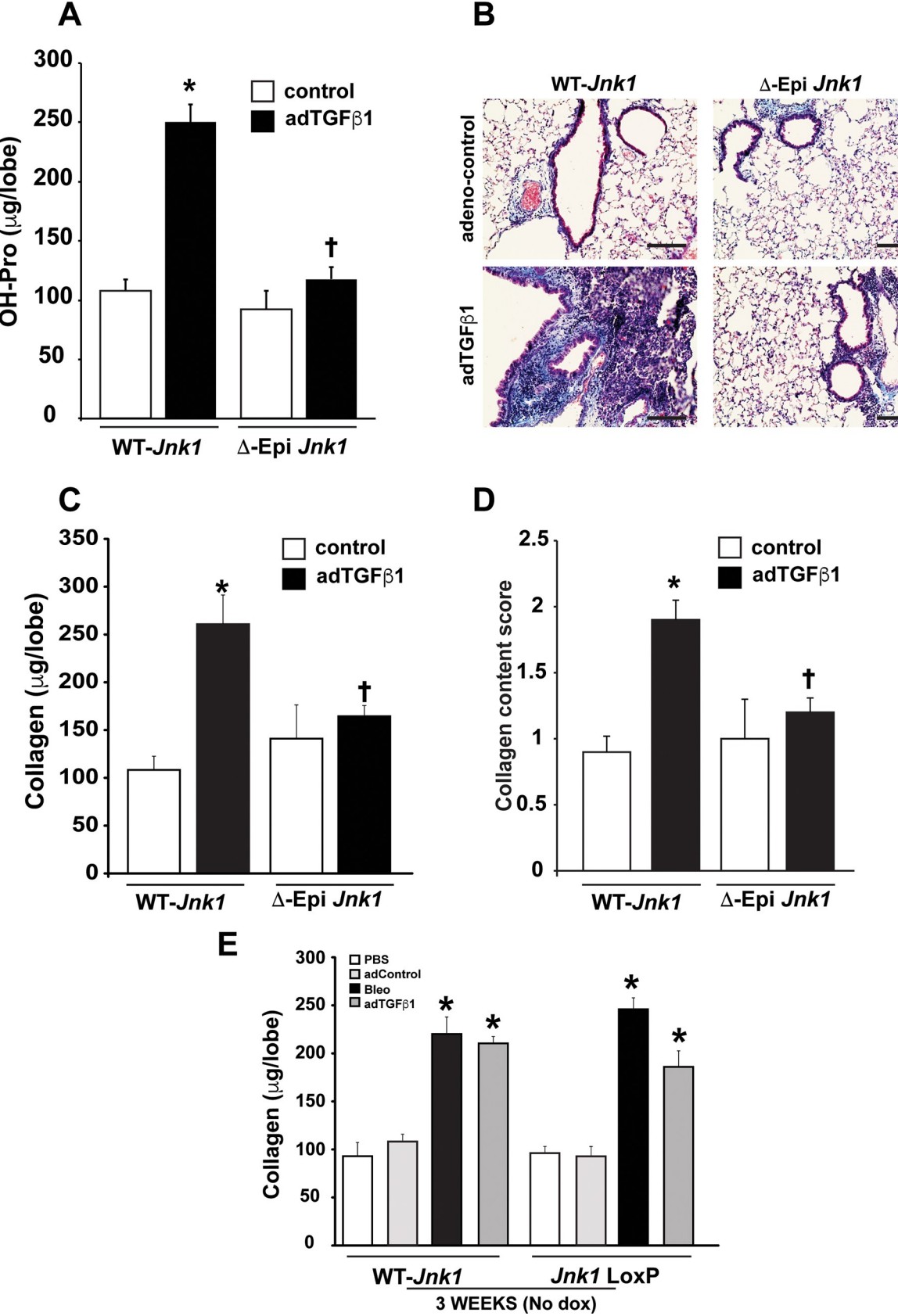

**Fig 4. Conditional ablation of JNK1 from lung epithelial cells attenuates lung fibrosis induced by adTGFβ1.** JNK1 was selectively ablated from the bronchiolar epithelial cells and type II cells using a CCSP promoter driving a reverse tetracycline activator (rtTA), the tetracyclin operon driving CRE recombinase (TetOP-Cre), and mice carrying a LoxP-flanked *Jnk1* allele. Assessment of total collagen content in lung tissue 3 weeks following administration of bleomycin via hydroxyproline content (**A**), Masson's trichrome staining (**B**), Sircol assay (**C**) and semi-quantitative evaluation of Masson's trichrome reactive material (**D**). **E:** Assessment of pulmonary fibrosis, using the sircol assay in response to AdTGFβ1 in mice containing the CCSP-rtTA, TetO-Cre, and *Jnk1* loxP/loxP alleles, or CCSP-rtTA, TetO-Cre alleles, in the absence of doxycycline-containing food. (WT- *Jnk1*: AdCtr n = 6, adTGFβ1 n = 8, ΔEpi *Jnk1*: AdCtr n = 5, adTGFβ1 n = 8 respectively mice/group from 2 independent experiments). * p< 0.05 compared to the adCtr control groups. † p< 0.05 compared to respective WT groups. (ANOVA). Scale bars: 50 μm.

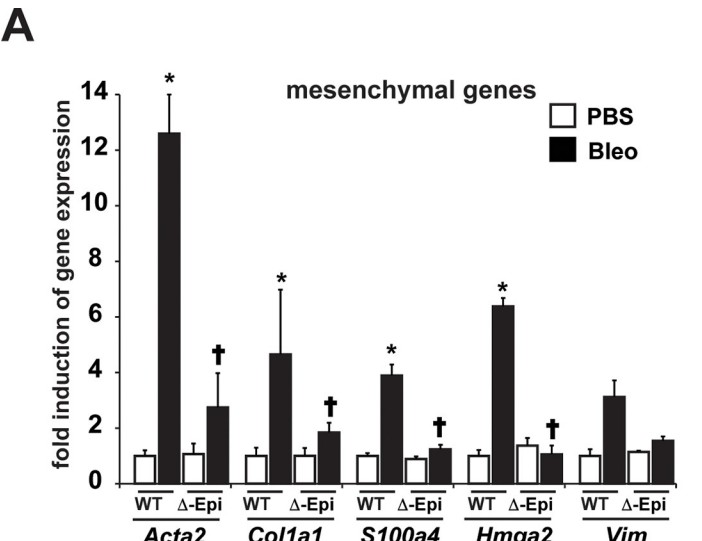

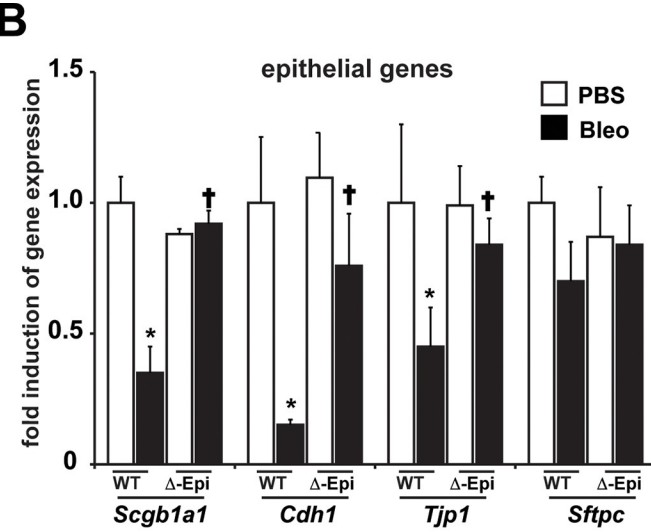

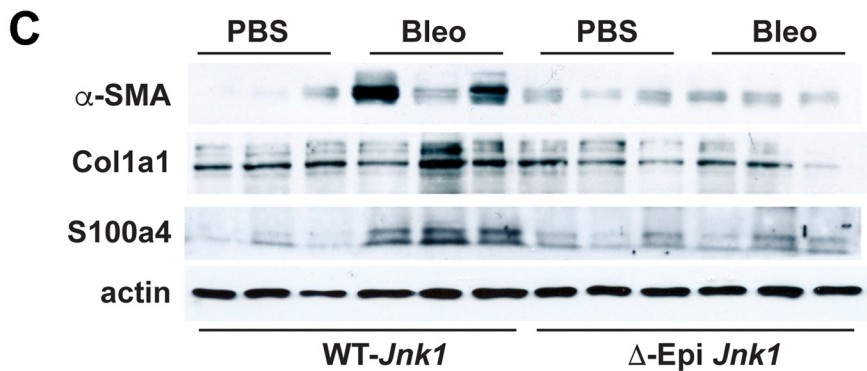

**Fig 5. Evaluation of epithelial and mesenchymal gene expression profiles in lungs from ΔEpi*Jnk1* mice, or control groups subjected to bleomycin-induced lung fibrosis.** Analysis of mesenchymal (**A**) or epithelial (**B**) mRNA expression in homogenized lung tissue from mice exposed to bleomycin for 3 weeks. Results were normalized to the housekeeping gene cyclophilin, and are expressed as fold expression changes (+/- SEM) compared to the WT vehicle control groups. (WT- *Jnk1*: PBS n = 6, Bleo n = 12, ΔEpi*Jnk1*: PBS n = 5, Bleo n = 8 respectively mice/group from 2 independent experiments). * p< 0.05 compared to the PBS control group. † p< 0.05 compared to the respective WT group. (ANOVA). **C:** Evaluation of mesenchymal proteins (α-SMA (*Acta2*), Col1a1, FSP1 (*S100a4*)) in lungs from ΔEpiJNK1 mice, or control groups subjected to bleomycin-induced lung fibrosis, and the impact of ablation of epithelial JNK1. Homogenized lung tissues were subjected to Western blot analysis for the indicated proteins. β-actin: Loading control. Shown are results from individual mice.

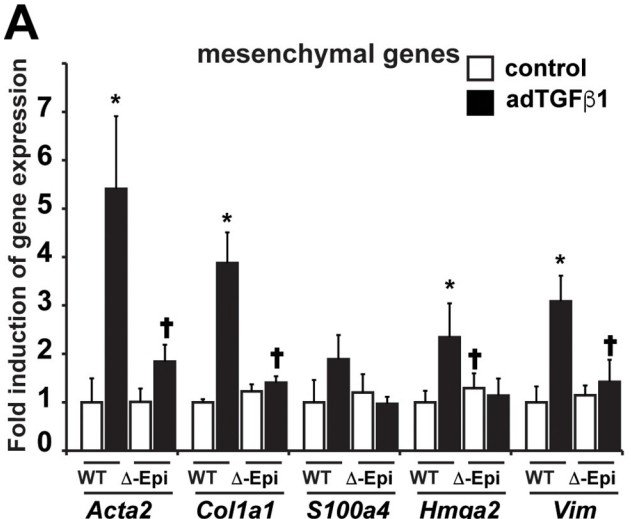

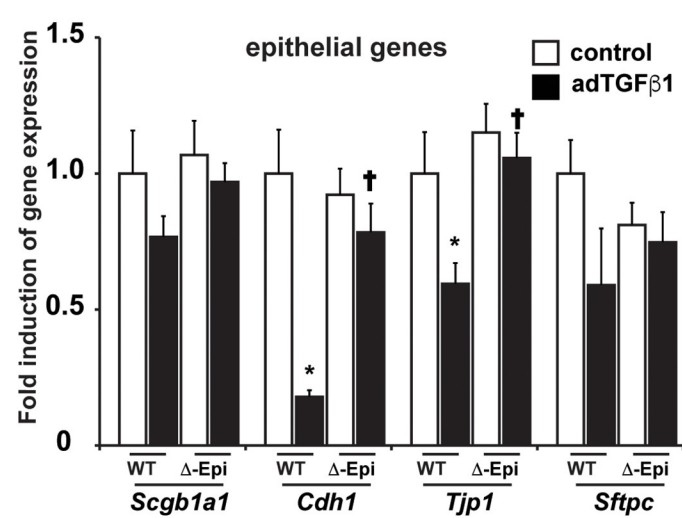

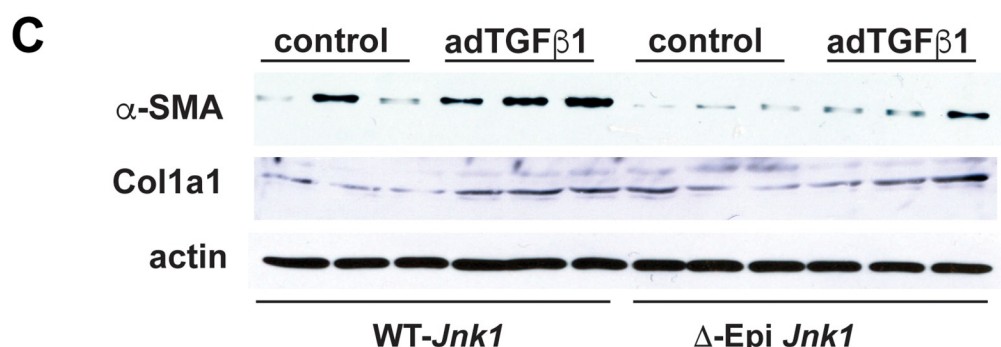

**Fig 6. Evaluation of epithelial and mesenchymal gene expression profiles in lungs from ΔEpi*JNK*1 mice, or control groups subjected to adTGFβ1-induced lung fibrosis.** Analysis of mesenchymal (**A**) or epithelial (**B**) mRNA expression in homogenized lung tissue from mice exposed to adTGFβ1 for 3 weeks. Results were normalized to the housekeeping gene cyclophilin, and are expressed as fold expression changes (+/- SEM) compared to the WT vehicle control groups. (WT- *Jnk1*: AdCtr n = 6, adTGFβ1 n = 8, ΔEpi *Jnk1*: AdCtr n = 5, adTGFβ1 n = 8 respectively mice/group from 2 independent experiments). * p< 0.05 compared to the adCtr control groups. † p< 0.05 compared to respective WT groups. (ANOVA). **C:** Evaluation of mesenchymal proteins (α-SMA (*Acta2*) and Col1a1 in lungs from ΔEpi *Jnk1* mice or control groups (WT mice) subjected to adTGFβ1-induced lung fibrosis. Homogenized lung tissues were subjected to Western blot analysis for the indicated proteins. β-actin: Loading control. Shown are results from individual mice.

observed when JNK1-deficient airway basal cells were plated onto lung scaffolds derived from patients with IPF. These findings suggest that JNK1 can act as a double-edged sword in the settings of fibrosis. JNK1 can regulate the fibrotic injury process via Smad and Wnt signaling pathways independent from the inflammatory responses. Conversely, JNK1 is important in the response of epithelial cells to an altered ECM in settings of pulmonary fibrosis and the subsequent loss of epithelial proteins. Additional studies will be required to identify the precise molecular targets of JNK1 that regulate the initial fibrotic response versus the ECM-derived cues and the relative importance of these events within epithelial cells in the pathogenesis of fibrosis.

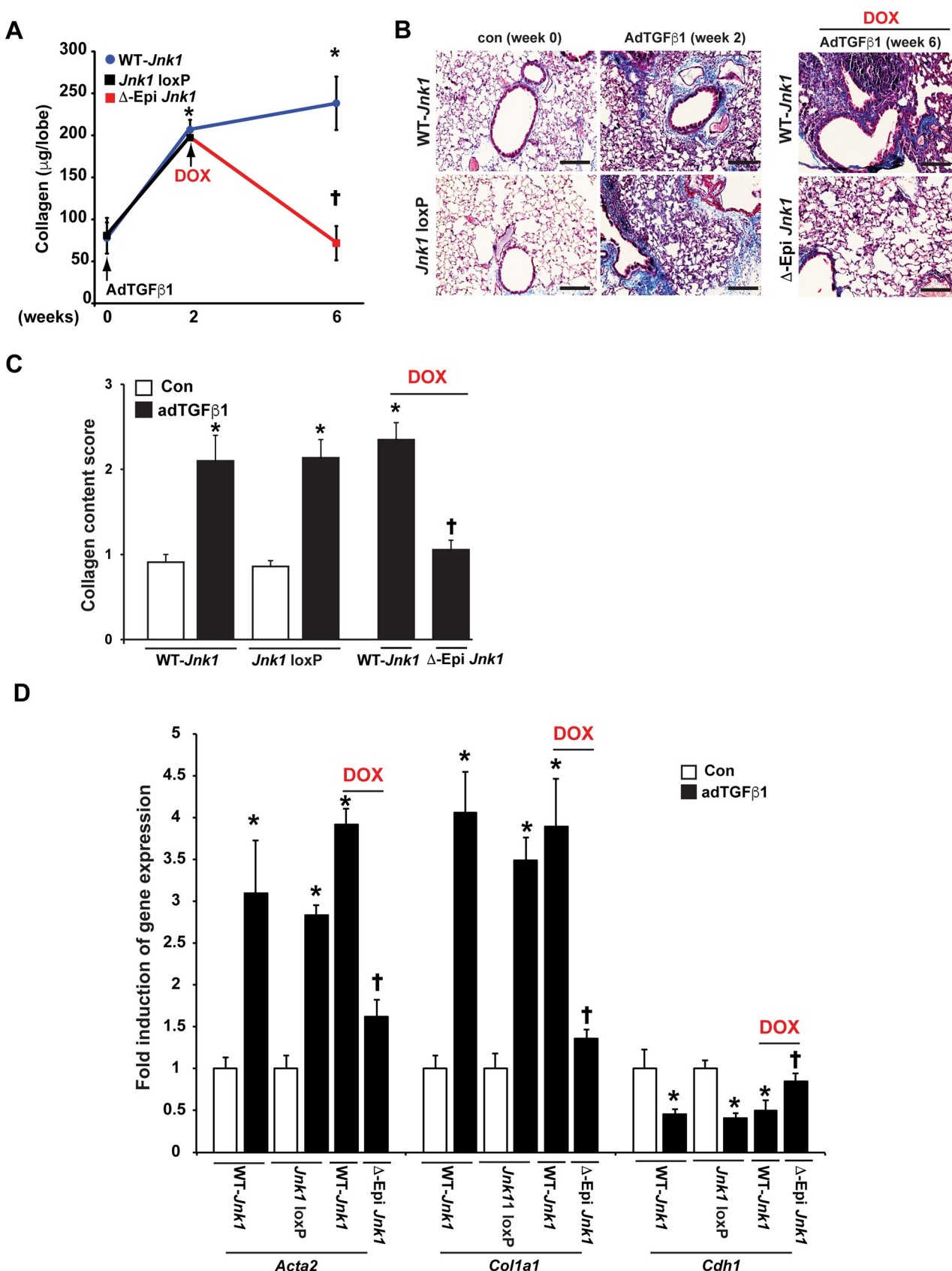

**Fig 7. The impact of delayed ablation of epithelial JNK1 in adTGFβ1-induced lung fibrosis.** Delayed ablation of JNK1 from lung epithelial cells reverses existing increases in lung fibrosis induced by AdTGFβ1. Mice expressing either CCSP-rtTA TetO-CRE transgenes along with the WT *Jnk1* allele, or CCSP-rtTA, tetO-CRE, *Jnk1*$^{LoxP/LoxP}$ transgenes (WT *Jnk1*) were exposed to AdTGFβ1 or AdCtr vector. As a control, CCSP-rtTA, tetO-CRE, *Jnk1*$^{LoxP/LoxP}$ transgene expressing mice were kept on regular chow (*Jnk1* LoxP). Two weeks thereafter, 3 mice/group were euthanized for assessment of total lung collagen content while the remainder of animals was administered dox food for an additional 4 weeks prior to assessment of total lung collagen content. Lung fibrosis was assessed via the Sircol assay (**A**) or Masson's Trichrome staining and analysis (**B and C**). **D**: Expression of epithelial and mesenchymal genes in homogenized lung tissue of various groups. (n = WT- *Jnk1* and *Jnk1* Loxp (no dox) adCtr/adTGFβ1 = 4/4/6/6, WT- *Jnk1* and ΔEpi *Jnk1* (on dox) adTGFβ1 = 8/11 respectively mice/group from 2 independent experiments). Results were normalized to the housekeeping gene cyclophilin, and are expressed as fold expression changes (+/- SEM) compared to the WT vehicle control groups. * p< 0.05 compared to PBS or AdCtr control groups. † p< 0.05 compared to respective WT groups. (ANOVA). Scale bars: 50 μm.

It had been demonstrated previously that the ECM is an important regulator of myofibroblast activation [32]. Notably, a fibrotic ECM scaffold was shown to lead to activation of control fibroblasts, while conversely, plating of activated myofibroblasts derived from IPF patients onto an ECM scaffold from control lungs resulted into a quiescent phenotype similar to control fibroblasts, suggesting that ECM is a prominent determinant that dictates fibroblast activation [33]. The activation of Rho kinase and myocardin-related transcription factor (MRTF) were shown to be critical mechano-sensing pathways that controlled myofibroblast activation [34]. Other studies have shown that the YAP/TAZ pathway also regulates the response of fibroblasts to ECM-derived cues [35]. The extent to which ECM regulates epithelial cell plasticity and the relative importance of the ECM in regulating epithelial cell plasticity or death will also require further study. Similarly, the role of JNK1 in ECM-controlled activation of epithelial cells, and the molecular targets of JNK herein also require further study. One putative target of JNK with relevance to mechano-transduction is the 14.3.3 protein, which can be phosphorylated on Ser 184 [36]. Further, there is compelling evidence for interplay between JNK and YAP/TAZ signaling pathways. JNK effector activator protein (AP)-1 activity is reportedly increased by YAP/TAZ through JNK–YAP interaction [37] and YAP/TAZ activation promotes inflammation and atherogenesis by enhancing JNK [38].

The demonstration that delayed ablation of *Jnk1* in epithelial cells prevented the progression of fibrosis, and more importantly reversed the existing increases in fibrosis in the Adeno TGFβ1 model of fibrosis indicates the potential therapeutic relevance of targeted inhibition of JNK1 in fibrosis. Indeed, in a recent study we showed that the JNK inhibitor, CC-930, attenuated bleomycin- and house dust mite- induced pulmonary fibrosis in mice [39, 40] although pharmacological JNK inhibition did not affect airway mechanics. In this study epithelial JNK1 ablation reduces lung elastance which is in line with the finding by Dolhnikoff et al. who observed that increased lung elastance in bleomycin-exposed animals correlated with collagen content [41], lending further support to the notion that modulating JNK might have a positive physiological effect in fibrotic lung disease. A clinical trial with CC-930 also showed some evidence of a potential clinical response in patients with dermal fibrosis and IPF [40, 42]. However, the latter trial had to be terminated due to side effects. A new clinical trial using a more specific JNK1 inhibitor, CC90001, is currently ongoing in patients with IPF (NCT03142191: clinical trials.gov). The demonstration of a strongly protective effect on ablation of epithelial JNK1 in fibrosis, linked to the concurrent clinical development of JNK1 inhibitors point to the potential clinical and translational relevance of our current findings.

## Supporting information

**S1 Fig. Schematics depicting mouse experiments.**
(TIF)

**S2 Fig. Raw western blot images.**
(PDF)

**S1 File. Excel data files.**
(XLSX)

## Acknowledgments

We would like to thank Dr. Jeffrey Whitsett, University of Cincinnati for generously providing respectively the CCSP-rtTA, TetOP-Cre mice, Dr. Jack Gauldie for providing AdTGFβ1, Dr. Barry Stripp for advice regarding lung cell fractionation and providing the CCSP antibody, Karolyn Lahue for assisting with the animal experiments and Minara Aliyeva and Nirav Daphtary for technical assistance measuring airway mechanics.

## Author Contributions

**Conceptualization:** John F. Alcorn, Yvonne M. W. Janssen-Heininger.

**Data curation:** Jos L. van der Velden, David G. Chapman.

**Formal analysis:** Jos L. van der Velden, David G. Chapman, Lennart K. A. Lundblad, Kelly Butnor.

**Funding acquisition:** Yvonne M. W. Janssen-Heininger.

**Investigation:** Jos L. van der Velden.

**Methodology:** Jos L. van der Velden, David G. Chapman.

**Resources:** Roger J. Davis.

**Supervision:** Charles G. Irvin, Yvonne M. W. Janssen-Heininger.

**Writing – original draft:** Jos L. van der Velden, Kelly Butnor, Yvonne M. W. Janssen-Heininger.

**Writing – review & editing:** Jos L. van der Velden, John F. Alcorn, David G. Chapman, Lennart K. A. Lundblad, Charles G. Irvin.

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
