## [Decision Letter · Decision Letter 0]

8 Oct 2019

PONE-D-19-21143

Airway epithelial specific deletion of Jun-N-terminal kinase 1 attenuates pulmonary fibrosis in two independent mouse models

PLOS ONE

Dear Dr. van der Velden,

Thank you for submitting your manuscript to PLOS ONE. After careful consideration, we feel that it has merit but does not fully meet PLOS ONE’s publication criteria as it currently stands. Therefore, we invite you to submit a revised version of the manuscript that addresses the points raised during the review process, in particular, additional information regarding materials and methods and more discussion about the potential mechanisms underlying epithelial JNK1 knockout in attenuating lung fibrosis.  

We would appreciate receiving your revised manuscript by Nov 22 2019 11:59PM. To enhance the reproducibility of your results, we recommend that if applicable you deposit your laboratory protocols in protocols.io, where a protocol can be assigned its own identifier (DOI) such that it can be cited independently in the future. For instructions see: http://journals.plos.org/plosone/s/submission-guidelines#loc-laboratory-protocols

We look forward to receiving your revised manuscript.

Kind regards,

Wei Shi

Academic Editor

PLOS ONE

Journal Requirements:

2. Thank you for including your ethics statement: The samples from IPF patients and non diseased controls were obtained from the National Heart Lung and Blood Institute-sponsored Lung Tissue Research Consortium (LTRC). The clinical data and specimens have been de-identified by the LTRC. LTRC protocols were approved by the institutional review boards at UVM

4. Please include a copy of Table 1 which you refer to in your text.

Additional Editor Comments (if provided):

Reviewers' comments:

Reviewer's Responses to Questions

**Comments to the Author**

1. Is the manuscript technically sound, and do the data support the conclusions?

Reviewer #1: Yes

Reviewer #2: Yes

2. Has the statistical analysis been performed appropriately and rigorously? 

Reviewer #1: Yes

Reviewer #2: Yes

3. Have the authors made all data underlying the findings in their manuscript fully available?

Reviewer #1: Yes

Reviewer #2: Yes

4. Is the manuscript presented in an intelligible fashion and written in standard English?

Reviewer #1: Yes

Reviewer #2: Yes

5. Review Comments to the Author

Reviewer #1: In materials and methods, chemical and antibodies, the concentration of actin antibody should be checked. 1:500 or 1:5000?

In human lung tissues, the authors should write how many patient tissues are studied.

The authors should identificate whether the patients studied are male or female.

The authors should write ethic committee approval number for clinic and all animal studies.

In phospho-JNK staining human lung tissues, the authors should write incubation time and degree of antibody. The method should write more detaily.

In animals, the authors should write male or female mice?

The start of experiment, administered substance day and end of experiment should be schematized.

In gene expression, the authors should explain to use protocol that taqman or sybr green?

In results and in figure 1, the authors Said that JNK1 (JNK1/2) and JNK2 (JNK2/1). These are not clearly. They should explain.

In figures, "the collegen content score" should be expressed instead of the score (trichrome).

"fold induction of gene expression" should be expressed instead of fold induction.

Reviewer #2: In this manuscript, the authors addressed the role of epithelial JNK1 activation in the pathogenesis of lung fibrosis using both the bleomycin and AdTGFbeta1 models. The authors demonstrated nicely that activation of JNK1 occurred in both models of lung fibrosis in mice as well as in human IPF, epithelial deletion of JNK1 protected mice from fibrosis in both models, and finally delayed epithelial deletion of JNK1 resulted in reversal of existing fibrotic response. The experiments were well-controlled, the inducible epithelial JNK1 deletion model was sufficiently described and verified, and the analyses of fibrosis were adequate and convincing. I only have one question/concern:

The fact that epithelial deletion of JNK1 offered protection against lung fibrosis induced by both bleomycin and AdTGFbeta1 suggests that JNK1 works downstream of or in concert with TGFbeta1 in the fibrotic process. The fibrotic response in the bleomycin model, however, occurs subsequent to epithelial injury and inflammation. What is the role of JNK1 activation in lung injury and inflammation induced by bleomycin? If epithelial JNK1 deletion alters the injury/inflammatory response induced by bleomycin, then its effects on fibrosis would be secondary.

6. PLOS authors have the option to publish the peer review history of their article (what does this mean?). If published, this will include your full peer review and any attached files.

Reviewer #1: No

Reviewer #2: No

---

## [Author Response · Author response to Decision Letter 0]

18 Nov 2019

Editor’s comments:

Comment 1. Please ensure that your manuscript meets PLOS ONE's style requirements, including those for file naming. 

Response 1: The manuscript style has been revised to the PLOS ONE requirements.

Comment 2: Thank you for including your ethics statement: The samples from IPF patients and non-diseased controls were obtained from the National Heart Lung and Blood Institute-sponsored Lung Tissue Research Consortium (LTRC). The clinical data and specimens have been de-identified by the LTRC. LTRC protocols were approved by the institutional review boards at UVM. Please amend your current ethics statement to include the full name of the ethics committee/institutional review board(s) that approved your specific study. Once you have amended this/these statement(s) in the Methods section of the manuscript, please add the same text to the “Ethics Statement” field of the submission form (via “Edit Submission”).

Response 2: The full name of the committee has been amended and the Ethics statement has been updated.

Comment 3: PLOS ONE now requires that authors provide the original uncropped and unadjusted images underlying all blot or gel results reported in a submission’s figures or Supporting Information files. This policy and the journal’s other requirements for blot/gel reporting and figure preparation are described in detail. In your cover letter, please note whether your blot/gel image data are in Supporting Information or posted at a public data repository, provide the repository URL if relevant, and provide specific details as to which raw blot/gel images, if any, are not available. Email us at plosone@plos.org if you have any questions.

Response 3: All the data files are uploaded in the supporting information and this is noted in the cover letter.

Comment 4: Please include a copy of Table 1 which you refer to in your text.

Response 4: Table 1 has been included in the manuscript.

Comment 5: Please include captions for your Supporting Information files at the end of your manuscript, and update any in-text citations to match accordingly. Please see our Supporting Information guidelines for more information: http://journals.plos.org/plosone/s/supporting-information.

Response 5: Captions have been included for the supporting information.

Reviewer comments #1: 

Comment1: In materials and methods, chemical and antibodies, the concentration of actin antibody should be checked. 1:500 or 1:5000?

Response 1: The concentration of the actin antibody is correct, 1:500.

Comment 2: In human lung tissues, the authors should write how many patient tissues are studied. The authors should identificate whether the patients studied are male or female.

Response 2: We agree with the reviewer that we should indicate the gender distribution of the analyzed samples. We included this in the methods section.

Comment 3: The authors should write ethic committee approval number for clinic and all animal studies.

Response 3: We included the Institutional Review Board Committee on Human Research in Medical Sciences approval number in the methods section. 

Comment 4: In phospho-JNK staining human lung tissues, the authors should write incubation time and degree of antibody. The method should write more detaily.

Response 4: We appreciate the reviewers comment and we included a more detailed protocol with regard to incubation time and antibody specifics in the methods.

Comment 5: In animals, the authors should write male or female mice?

Response 5: we agree with the reviewer and included a statement that we use both male and female mice.

Comment 6: The start of experiment, administered substance day and end of experiment should be schematized.

Response 6: We included experimental schematics in supplemental figure 1

Comment 7: In gene expression, the authors should explain to use protocol that taqman or sybr green?

Response 7: We indicated that we used SYBR green.

Comment 8: In results and in figure 1, the authors Said that JNK1 (JNK1/2) and JNK2 (JNK2/1). These are not clearly. They should explain.

Response 8: Our apologies this is not clear, in the methods section we clearly explain why it is labeled this way together with a reference. In order to prevent confusion, we changed the labeling of figure 1A. 

Comment 9: In figures, "the collegen content score" should be expressed instead of the score (trichrome). "fold induction of gene expression" should be expressed instead of fold induction.

Response 9: We appreciate the reviewer comments and corrected the figures.

Reviewer comments #2

Comment 1: The fact that epithelial deletion of JNK1 offered protection against lung fibrosis induced by both bleomycin and AdTGFbeta1 suggests that JNK1 works downstream of or in concert with TGFbeta1 in the fibrotic process. The fibrotic response in the bleomycin model, however, occurs subsequent to epithelial injury and inflammation. What is the role of JNK1 activation in lung injury and inflammation induced by bleomycin? If epithelial JNK1 deletion alters the injury/inflammatory response induced by bleomycin, then its effects on fibrosis would be secondary.

Response 1: We agree with the reviewer that this is an interesting dilemma and appreciate his comments. In previous mouse studies; both in bleomycin-induced fibrosis and house dust mite-induced allergic airway disease models we have not observed a significant impact of JNK1 on the inflammatory responses. A statement of this finding has been included in the conclusion. Further, we also have shown that JNK1 cannot only regulate the injury process via Smad and Wnt signaling pathways (reference 16 and 17) but JNK1 also plays a role in the “epithelial-ness” of cells (reference 19). In this particular study we show that JNK1 is important in the response of epithelial cells to a “diseased” extracellular matrix. These two findings combined suggest that JNK1 can act as a double edge sword. On one hand it can regulate the severity of the fibrotic injury response potentially via Smad and Wnt signaling pathways and on the other hand JNK1 can play a role in the repair process via regulating re-epithelialization. We added this to the discussion.

---

## [Decision Letter · Decision Letter 1]

10 Dec 2019

Airway epithelial specific deletion of Jun-N-terminal kinase 1 attenuates pulmonary fibrosis in two independent mouse models

PONE-D-19-21143R1

Dear Dr. van der Velden,

We are pleased to inform you that your manuscript has been judged scientifically suitable for publication and will be formally accepted for publication once it complies with all outstanding technical requirements.

With kind regards,

Wei Shi

Academic Editor

PLOS ONE

Additional Editor Comments (optional):

Reviewers' comments:

Reviewer's Responses to Questions

**Comments to the Author**

1. If the authors have adequately addressed your comments raised in a previous round of review and you feel that this manuscript is now acceptable for publication, you may indicate that here to bypass the “Comments to the Author” section, enter your conflict of interest statement in the “Confidential to Editor” section, and submit your "Accept" recommendation.

Reviewer #2: All comments have been addressed

2. Is the manuscript technically sound, and do the data support the conclusions?

Reviewer #2: Yes

3. Has the statistical analysis been performed appropriately and rigorously? 

Reviewer #2: Yes

4. Have the authors made all data underlying the findings in their manuscript fully available?

Reviewer #2: Yes

5. Is the manuscript presented in an intelligible fashion and written in standard English?

Reviewer #2: Yes

6. Review Comments to the Author

Reviewer #2: (No Response)

7. PLOS authors have the option to publish the peer review history of their article (what does this mean?). If published, this will include your full peer review and any attached files.

Reviewer #2: No

---

## [Editor Report · Acceptance letter]

16 Dec 2019

PONE-D-19-21143R1 

Airway epithelial specific deletion of Jun-N-terminal kinase 1 attenuates pulmonary fibrosis in two independent mouse models 

Dear Dr. van der Velden:

I am pleased to inform you that your manuscript has been deemed suitable for publication in PLOS ONE. Congratulations! Your manuscript is now with our production department. 

With kind regards,

on behalf of

Dr. Wei Shi 

Academic Editor

PLOS ONE